# Assessment of Risk Factors for Development of Overweight and Obesity among Soldiers of Polish Armed Forces Participating in the National Health Programme 2016–2020

**DOI:** 10.3390/ijerph19053069

**Published:** 2022-03-05

**Authors:** Agata Gaździńska, Paweł Jagielski, Marta Turczyńska, Łukasz Dziuda, Stefan Gaździński

**Affiliations:** 1Laboratory of Dietetics and Obesity Treatment, Department of Psychophysiological Measurements and Human Factor Research, Military Institute of Aviation Medicine, Krasińskiego 54/56, 01-755 Warsaw, Poland; mturczyn@wiml.waw.pl; 2Department of Nutrition and Drug Research, Institute of Public Health, Faculty of Health Sciences, Jagiellonian University Medical College, 31-066 Krakow, Poland; paweljan.jagielski@uj.edu.pl; 3Department of Psychophysiological Measurements and Human Factor Research, Military Institute of Aviation Medicine, Krasińskiego 54/56, 01-755 Warsaw, Poland; ldziuda@wiml.waw.pl; 4Department of Neuroimaging, Military Institute of Aviation Medicine, Krasińskiego 54/56, 01-755 Warsaw, Poland; stefan.gazdzinski@yahoo.com

**Keywords:** BMI, body mass index, obesity, overweight, risk factors, soldiers

## Abstract

The aim of this study was to assess the prevalence of overweight and obesity among Polish Armed Forces soldiers and to analyze risk factors impacting body mass. In total, 1096 male, Caucasian soldiers (36.31 ± 8.03 years) participated in this study. Anthropometric data were obtained, and questionnaires evaluated sociodemographic, environmental, behavioral and biological factors known to be associated with obesity. Only 33% of the total number of participants had normal body weight, and 17.3% were considered obese (according to WHO criteria). The results showed that being 40 years or older, sleeping six hours or less per day, more frequent reaching for food in stressful situations, having a mother with excessive body weight, not exercising or exercising at most two days per week, and spending two hours a day or more in front of the TV increase the risk of obesity. Taken together, the results show that factors such as family history of obesity, dietary habits, physical activity, length of sleep and ability to cope with stress could be used to identify soldiers at higher risk of developing obesity in order to provide them with personalized prevention programs.

## 1. Introduction

Epidemiological data indicate that overweight and obesity are two of the main public health problems. They affect all age groups and countries all over the world regardless of their developmental stage [1,2,3,4]. According to a report by the World Health Organization (WHO), the prevalence of overweight and obesity has increased dramatically in the general population over the past decades. Now, 39% of adults worldwide are overweight and 13% are obese ([5] (https://www.who.int/news-room/fact-sheets/detail/obesity-and-overweight, accessed on 27 August 2021)).

It is now widely accepted that occupational factors may play an important role in the occurrence of excessive body weight [6,7]. Military personnel, as an occupational group, are more exposed to stress, harmful environmental factors and limitations in food selection or availability, especially during military exercises or military missions [8,9].

Studies have shown an alarmingly high prevalence of excessive body weight among soldiers all over the world. Soldiers fulfilling the criteria of obesity constitute 8% in the US Army [10], 12% in the British Army [11], 13% in the Iranian Army [12,13] and 6% in the Polish Air Force [14]. A higher percentage of soldiers with excessive body weight was found in the Saudi Arabian Army, where obesity was diagnosed in 44% of soldiers [15].

Alarmingly high percentages of overweight and obese Polish soldiers, 58% and 27%, respectively, in those over 50 years of age were reported by Gielerak et al. in the implementation of the MIL-SCORE (Equalization of Accessibility to Cardiology Prophylaxis and Care for Professional Soldiers) program, which aimed to assess the prevalence of cardiovascular risk factors in a population of 6440 Polish soldiers [16]. According to the authors of the program, the professional staff of the Polish Army, due to the specific conditions of performing their tasks, seems to be particularly exposed to risk factors for adverse cardiovascular events. The results of the MIL-SCORE program showed the occurrence of hypertension in 45% of soldiers and lipid disorders in more than half of the participants. Coronary heart disease was diagnosed in 3% of Polish soldiers over 50 years of age and <1% of younger soldiers. Diabetes affected 7% of the oldest soldiers and 3% of soldiers aged 40–50 years. In the age subgroup > 50 years, high and very high results of cardiovascular risk assessment were observed in almost one-third of soldiers [16]. The results of the MIL-SCORE program indicate that Polish soldiers have multiple cardiovascular risk factors, including obesity, which reflect trends observed in the general population. This confirms the trends observed in earlier studies [17].

The above studies on obesity among military personnel most often evaluated sociodemographic factors such as age, sex, marital status, military rank and employment. Only one of these studies evaluated the effects of fitness and physical activity on body weight [13]. Risk factors contributing to the obesity epidemic are of interest to many researchers worldwide. In particular, the Organisation for Economic Cooperation and Development (OECD) identifies economic, social and physical factors that have changed dynamically over the past 20–30 years and have influenced diets and levels of physical activity at work, at home and leisure time [18]. Other researchers have placed emphasis on lifestyle factors [19], including particular attention to the impact of stress on weight gain [20,21].

In the present work, as compared to previous studies on military personnel, we propose considering the simultaneous influence of a much larger number of environmental and behavioral factors on the prevalence of obesity among Polish soldiers. These factors include time spent in front of the television [22,23], time spent in front of the computer [24], time spent sleeping [25], dietary factors (e.g., snacking, consumption of fast food and sweets) [26], family history of obesity [22], leisure time activities [27] and the effects of mood and stress on eating behavior [28].

To our knowledge, despite the constantly deteriorating nutritional status among soldiers of the Polish Armed Forces, no one has undertaken an investigation of the risk factors for overweight and obesity in this professional group so far. In addition to assessing the most important risk factors for the development of overweight and obesity among soldiers of the Polish Armed Forces, the aim of our work was to propose a model of the probability of obesity based on risk factors. Achieving these objectives will help design more effective obesity prevention programs for these soldiers.

## 2. Materials and Methods

### 2.1. Participants

Full anthropometric and questionnaire data were obtained in a group of 1096 Caucasian men (36.31 ± 8.03 years, mean body mass 84.94 ± 14.84 kg, mean body height 177.56 ± 7.35 cm and mean body mass index (BMI) 26.85 ± 3.95), soldiers of the Polish Armed Forces (370 soldiers of the Navy, 283 soldiers of the Land Forces, 184 soldiers of the Air Force, 86 soldiers of the Garrison Warsaw and 41 soldiers of the Territorial Defence Forces) from military units from all over Poland that have declared their willingness to participate in the National Health Programme 2016–2020. The invitation to participate in the National Health Programme was earlier sent to all commanders of the military units. If they agreed, invitations had been sent via e-mail to their subordinates. All soldiers who expressed their willingness to participate in the Program were enrolled. The research was conducted between April 2018 and October 2020. An additional 133 participants had incomplete datasets and were not included in the analyses. All participants accounted for about 1% of the Polish Armed Forces.

All procedures were approved by the Institutional Review Board of the Military Institute of Aviation Medicine, Warsaw, Poland (decision No. 01/2018 of 9 March 2018), and have been performed in accordance with the ethical standards as laid down in the 1964 Declaration of Helsinki and its later amendments or comparable ethical standards. All participants provided informed consent.

### 2.2. Study Design

Body height was measured with an anthropometer (Holtain, UK, https://holtain.co.uk/anth.php (accessed on 20 January 2022) to the nearest 1 mm, in a standing upright position, without shoes. Body weight was determined in underwear alone, after emptying the bladder. Participants were categorized on the basis of BMI based on World Health Organisation (WHO) criteria (5). Three groups were created: BMI in the range 18.5–24.9 (normal), BMI in the range 25.0–29.9 (overweight) and BMI ≥ 30.0 (obese). Participants with a BMI < 18.5 were not included.

### 2.3. Instruments for Data Collection

A research questionnaire was used to identify obesity risk factors. The questionnaire asked about age; education; size of the community; being obese as a child/teenager; body weight at the age of 18; family history of obesity; number of meals consumed and their size; feeling satiated after eating; snacking during the day and at night; time of eating the last meal before bedtime; consumption of sweets, fast-food products, sugar, salt and water; eating in front of the TV/computer; paying attention to the caloric value of meals and food labels; number of days during the last week (7 days) when the respondent did moderate physical exercise; time spent in front of the TV; length of sleep; and ability to cope with stress. Respondents completed the survey in an online version. All procedures were performed in the morning hours.

### 2.4. Statistical Analysis

Descriptive statistics were calculated: mean, standard deviation and median. Compliance with the normal distribution of quantitative variables was checked using the Shapiro–Wilk test. The Kruskal–Wallis analysis of variance was used to check the differences between the three groups (depending on the BMI categories). A chi-square test was used to test for differences between participants’ BMI categories and obesity risk factors. The multivariable logistic regression was applied to calculate odds ratios (ORs) and 95% confidence intervals (CIs). Statistical analyses were performed using PS IMAGO PRO 7 (IBM SPSS Statistics 27, Armonk, NY, USA). The level of statistical significance was set at *p* < 0.05.

### 2.5. Logistic Regression Model

Due to the fact that the effect of increased BMI may be related to expanded muscle mass (so-called muscular overweight), which is relatively common among soldiers [29,30,31], only groups with BMI corresponding to normal body weight and obesity were used to determine the odds ratio (OR) of BMI ≥ 30 for the risk factors in question. Overweight participants were not included in the analyses, as this group might have included participants with normal body fat and increased muscle mass. Logistic regression was used to calculate the probability of obesity in the study group of professional soldiers, depending on risk factors and their combinations. OR was calculated for factors that were found to be significant in initial verification using the chi-square test. The OR indicates how many times the risk of obesity (BMI ≥ 30) is higher in the group of participants with a given risk factor compared to the group of participants without the same risk factor. The model of the probability of obesity among study soldiers based on statistically significant risk factors is presented below.
$$P\;\left(Y=1\right)=\frac{e^{-1.977\;+\;0.994*STRESS\;+\;0.968*MOTHER+0.599*PHISICAL\;ACTIVITY\;+\;1.315*AGE}}{1\;+\;e^{-1.977\;+\;0.994*STRESS\;+\;0.968*MOTHER+0.599*PHISICAL\;ACTIVITY\;+\;1.315*AGE}}$$

*P*—probability;*e*—base of natural logarithm;*STRESS*—eating when experiencing negative emotions (0 = No, 1 = Yes);*MOTHER*—excessive maternal weight (0 = No, 1 = Yes);*PHYSICAL ACTIVITY*—moderate exercise less than 3 days per week (0 = No, 1 = Yes);*AGE*—age over 40 years (0 = No, 1 = Yes).

Based on the equation presented, the probability of obesity in the absence of risk factors as well as in the simultaneous presence of four risk factors was calculated.

## 3. Results

The mean age of the respondents participating in the study was 36.31 ± 8.03 years. The mean BMI of all participants was 26.85 ± 3.95. Only 33% of the total number of participants had normal body weight, 49.6% were overweight and 17.3% were considered obese (according to WHO criteria). The mean values per BMI group are presented in Table 1.

The soldiers at normal body weight were younger (33.78 ± 7.59 years) than soldiers with diagnosed obesity (39.93 ± 7.71 years). Most (50.7%) of the respondents lived in a city of more than 100 thousand inhabitants, 30.9% lived in a city of fewer than 100 thousand inhabitants and 18.4% lived in the countryside; 55.4% of the respondents were persons with higher education, and the rest of the soldiers had secondary education. Residence and education level of the participants had no significant effect on their body weight (*p* < 0.05).

The soldiers belonging to respective BMI groups showed no significant differences in the reported number of consumed meals per day, snacking between main meals, snacking at night, time of last meal consumption, salt intake, fast-food consumption, amount of consumed water, paying attention to the calorie content of meals, reading food labels or eating in front of TV. The results are presented in Table 2. Participants with BMI ≥ 30 reported being significantly more likely to be overfed after meals (*p* = 0.016) and to reach for food when experiencing negative emotions and bad mood as compared to participants with normal BMI (28.9% vs. 10.8%) (*p* = 0.0001). Soldiers with BMI ≥ 30 were significantly more likely to sweeten beverages as compared to soldiers with a normal body weight (47.4% vs. 57.7%) (*p* = 0.002).

The analysis of the physical activity data showed that participants with BMI ≥ 30 performed moderate physical activity less frequently (*p* < 0.001) in the week preceding the study (Table 3), while they spent significantly more time in front of a computer and TV compared to normal weight and overweight participants (*p* = 0.003). Participants with obesity spent on average 4.31 ± 3.41 h/day in front of the computer and 1.79 ± 1.31 h/day in front of the TV.

In response to the question concerning body weight in childhood, most of the respondents declared that they had a normal body weight (76.3%), 11.4% declared that they were overweight, 2.3% declared that they were obese and 10.1% declared that they were underweight (Table 4). Soldiers with BMI in the obese range were significantly more likely to report excessive body weight in childhood (*p* < 0.001). In response to the question regarding body weight at age 18, 82.2% declared that they had a normal body weight, 8.7% declared that they were overweight, 1.3% declared that they were obese and 7.8% declared that they were underweight.

Regarding the prevalence of obesity in parents, 62.7% of the soldiers at normal weight declared that their mothers were or had been of normal weight and 66.7% that their fathers were or had been of normal weight. Soldiers with obesity were significantly more likely to declare that their parents were overweight or obese (*p* < 0.001) as compared to soldiers at normal weight. Detailed results are presented in Table 4.

Interestingly, salting, eating sweets, eating fast food, snacking between meals, the time of the last meal before bedtime and snacking at night did not differ between soldiers belonging to the three groups (Table 2 above).

### Risk Factors for BMI ≥ 30

The results of the analysis showed that six or less hours of sleep per day increased the risk of obesity by 74% (OR = 1.74; 95% CI: 1.19–2.54) in the study group with respect to those who slept seven or more hours (Table 5). More frequent reaching for food when experiencing negative emotions and bad mood and well-being increased the risk of BMI in the obese range by almost 4 times (OR = 3.64; 95% CI: 2.12–6.22) in relation to participants who declared they forgot to eat at that time or these situations did not influence their eating behavior.

Soldiers whose mothers were overweight or obese were almost 3 times more likely to develop obesity (OR = 2.86; 95% CI: 1.98–4.14) than soldiers whose mothers’ weight was normal. In contrast, those whose fathers were overweight or obese were almost twice more likely to develop obesity (OR = 1.78; 95% CI: 1.23–2.57).

Participants who did not exercise or exercised at most two days per week had more than 2 times the risk of obesity relative to those who reported exercising at a moderate level at least three days per week (OR = 2.28; 95% CI: 1.59–3.27). Participants who spent two or more hours a day in front of the TV had a 54% higher risk of obesity as compared to those who did not watch TV at all or watched no more than 2 h a day (OR = 1.54; 95% CI: 1.07–2.23).

Soldiers aged 40 years or older were more than 4 times more likely to develop obesity as compared to their younger counterparts (OR = 4.46; 95% CI: 3.04–6.55) (Table 5).

In the next analysis, the risk factors for obesity in the studied group of soldiers shown in Table 5 were entered into a logistic regression model. On the basis of multivariate analysis, the following factors remained statistically significant: reaching for food in situations of experiencing negative emotions and bad mood and well-being, excessive maternal body weight, practicing moderate-intensity physical activity less than 3 days a week and age over 40 years. Detailed results of the obtained logistic regression model are presented in Table 6. In the absence of risk factors, the probability of obesity in the studied group of soldiers was 12.2%, while in the case of the simultaneous occurrence of four risk factors, the risk of obesity was up to 86.9%.

## 4. Discussion

The emergence of obesity as a distinct disease could have far-reaching consequences for an organization where optimum health and physical fitness are required for personnel to perform their occupational roles effectively [32]. Our survey found a 17% prevalence of obesity in the Polish Armed Forces. This percentage is higher than that in the American, French, English or Iranian armies [10,11,12,13] but lower than that in the Saudi Army [15].

According to a report published in 2021 by the National Institute of Public Health–National Research Institute, in Poland, overweight occurs in 52.4% of men and 32% of women, and obesity occurs in 16.5% and 16.2%, respectively. The prevalence of overweight and obesity in the Polish military is comparable to that in the general male population in Poland ([33] (Accessed on 20 January 2022).

As in our study, age was also one of the most significant risk factors for obesity among soldiers in the Saudi Arabian [34], American [10,35], French [13] and UK armed forces [11,36]. The fact that obesity increases with the age of soldiers is worrying, as it suggests that obesity develops while serving in the Armed Forces, contrary to our expectations [37].

Smith et al. [35] note that age was associated with obesity among military personnel, most likely as a result of gaining small amounts of weight over many years. We agree with the thesis of Quertier et al. [13], that this relationship could result from changes in employment type, such as from active operational roles to more sedentary posts, which induce weight gain.

We observed in the study group that the probability of obesity among soldiers who had an obese mother is 26%, which is consistent with the literature [38]. However, in our case, not only obesity of mothers but also obesity of fathers was associated with a higher probability of BMI ≥ 30 in soldiers [39,40]. According to many authors [22,23], children of obese parents prefer a sedentary lifestyle, are more likely to consume high-calorie foods and spend more time in front of the television.

Reaching for food when experiencing negative emotions and a bad mood proved to be another highly significant risk factor for the development of obesity in the studied cohort of soldiers. Stress is significantly related to nutrition [28]. It can alter overall food intake in two ways, causing malnutrition or overeating, which can be influenced by the severity of the stressor [21]. People seek to minimize the feelings of tension that accompany stressful situations by using the means available to them. Since food is often readily available and an immediately effective way of discharging emotions, it is often used to alleviate the effects of stress. Evidence from longitudinal studies suggests that chronic life stress may be causally linked to weight gain [21], consistent with our results. Studies by other authors also confirm that chronic occupational stress and a lack of social support at work are risk factors for obesity [20,41,42]. Therefore, it seems important to create programs to reduce stress among professional soldiers.

A modifiable lifestyle factor strongly associated with obesity among soldiers was also time spent sleeping and watching TV per day. Too much time watching television is also a lifestyle component associated with obesity, due to lack of exercise and/or frequent snacking promoted by TV watching [43]. Our results are consistent with previous studies showing an association between overall time spent in front of a screen and obesity, demonstrating that reducing time spent in front of a screen can reduce the prevalence of overweight and obesity [44]. Thus, the reduction of screen time should be considered for weight management programs.

There is consistent evidence that the amount of sedentary behavior (or low physical activity) is associated with higher levels of body fat [45]. In the military community, as in the general population, too little physical activity is also cited as one of the risk factors for obesity [13]. In this study, the probability of obesity in soldiers who spent less than three days per week on moderate exercise was 20%. In our study, 23.4% of soldiers who reported moderate physical activity up to two days during the last week had obesity. In the group of respondents who declared performing physical activity at least 3 days per week, 12% had obesity. In comparison, in Quartier’s study [13], 16% of servicemen who reported less than 2 h of physical activity were obese, while the obesity prevalence was only 8% among those practicing 2 to 4 h per week and 6.6% for members undertaking at least 4 h a week.

Sleep is one of the main dependent factors, besides physical activity and diet, leading to the maintenance of good health and thus is important among soldiers. Too little sleep causes metabolic and endocrine changes, including decreased glucose tolerance, decreased insulin sensitivity, increased evening cortisol levels, increased ghrelin levels, decreased leptin levels and increased hunger and appetite [46]. Epidemiological studies have found a significant association between short sleep (typically < six hours per night) and increased risk of obesity [44]. A meta-analysis of 18 studies involving 604,509 adults found a pooled odds ratio (OR) for obesity of 1.55 (1.43–1.68) for less than five hours of sleep [25]. It was also estimated that for each additional hour of sleep, BMI decreased by 0.35. Our own research found that six or fewer hours of sleep per day increased the risk of obesity by 74% among soldiers.

Analysis of the study results showed that not all risk factors for obesity present in the general population were significant in this study group, such as eating sweets, eating fast food, snacking between meals or snacking at night.

### Limitations

The selection of personnel was based on availability rather than on any particular characteristic associated with obesity. BMI alone may overdiagnose obesity because it does not take into account the real muscle mass, which is larger in soldiers than in the general population [29,31]. However, we used BMI to divide participants into groups to be consistent with the literature and to allow comparisons. It should be noted that soldiers in the obese range, but not in the overweight range, have an increased amount of body fat (e.g., [31]). Therefore, in the majority of our analyses, we compared soldiers at normal weight with soldiers with BMI exceeding 30 to minimize this bias. Nonetheless, our results are consistent with reports originating from other armies, as well as with general knowledge about obesity and its causes.

## 5. Conclusions

Overweight and obesity prevalence in the Polish Armed Forces is similar to that in other armies of developed countries and in the general population of Poland. Analyses of the results of the National Health Programme implemented in 2018–2020 clearly indicate that the level of existing efforts to prevent overweight and obesity among professional soldiers in Poland is insufficient. The results show that factors such as family history of obesity, dietary habits, physical activity, length of sleep and ability to cope with stress could be used to identify soldiers at higher risk of developing obesity in order to provide them with personalized prevention programs. Therefore, therapeutic interventions to reduce the prevalence of overweight and obesity among Polish soldiers should include self-monitoring of body weight especially after the age of 40, nutritional education, physical activity at least three days a week and psychological support for individuals who eat when experiencing negative emotions.

## Figures and Tables

**Table 1 ijerph-19-03069-t001:** Basic anthropometric parameters of the soldiers in relation to BMI.

Variable	All *n* = 1096		BMI 18.5–24.9 *n* = 362		BMI 25–29.9 *n* = 544		BMI ≥ 30 *n* = 190		
	X (SD)	Me	X (SD)	Me	X (SD)	Me	X (SD)	Me	*P*
Age (years)	36.31 (8.03)	36.00	33.78 (7.59)	33.00	36.71 (7.87)	36.00	39.93 (7.71)	40.00	**<0.001**
Body weight (kg)	84.94 (14.84)	84.70	71.81 (9.18)	73.00	86.74 (7.65)	86.00	104.82 (14.51)	102.00	**<0.001**
Height (cm)	177.56 (7.35)	178.00	175.80 (8.22)	176.50	178.59 (6.70)	179.00	177.96 (6.76)	178.00	**<0.001**
BMI	26.85 (3.95)	26.27	23.14 (1.49)	23.62	27.16 (1.39)	27.02	33.06 (3.90)	32.11	**<0.001**

*n*—sample size, X—mean value, SD—standard deviation, Me—median, *p*—*p* value for Kruskal–Wallis test result. Bold values denote statistical significance at the *p* < 0.05 level.

**Table 2 ijerph-19-03069-t002:** Eating behavior of soldiers according to BMI (*n* = 1096).

Question	Response	All (%)	BMI 18.5–24.9 (%)	BMI 25–29.9 (%)	BMI ≥ 30 (%)	*p*
Do you snack between main meals?	Always	7.6	8.6	6.8	8.0	0.585
	Sometimes	40.3	42.6	39.4	38.5	
	Never	52.1	48.7	53.8	53.5	
What kind of food do you eat most frequently between meals?	Sweets	18.5	18.1	19.1	17.3	0.042
	Fruit	31.4	36.2	30.8	24.0	
	Vegetables	8.1	9.2	7.4	8.4	
	Sandwiches	11.4	7.5	12.9	14.5	
	Everything	30.6	29.0	29.9	35.8	
How do you usually feel after meals?	Insatiable	7.0	6.7	6.5	9.0	0.016
	Stuffed	86.7	89.2	87.4	79.9	
	Overfed	6.3	4.2	6.1	11.1	
How many hours before bedtime do you eat your last meal?	Just before bedtime	3.6	3.3	3.3	4.7	0.209
	1 h before bedtime	20.7	17.7	21.5	24.2	
	2–3 h before bedtime	65.8	68.7	64.2	65.3	
	4 or more hours before bedtime	9.9	10.2	11.0	5.8	
Have you ever snacked at night?	No	80.6	83.1	80.5	76.2	0.193
	Yes, several times a week	3.3	1.7	4.0	4.2	
	Yes, several times a month	7.4	6.4	7.9	7.9	
	Yes, less than once a month	8.7	8.8	7.5	11.6	
How many hours do you sleep most often?	Less than 6 h	16.9	12.7	17.2	24.2	**0.003**
	About 6 h	47.6	46.4	48.5	47.4	
	7–8 h	34.6	40.6	33.0	27.4	
	More than 8 h	0.9	0.3	1.3	1.1	
When experiencing negative emotions, bad mood and well-being…	I reach for food more often	14.0	10.8	10.9	28.9	**< 0.001**
	I forget to eat	24.9	25.8	26.3	18.9	
	This has no effect on my eating behavior	61.2	63.4	62.8	52.1	
Do you sweeten your beverages/food?	No	52.4	51.1	56.8	42.3	**0.002**
	Yes	47.6	48.9	43.2	57.7	
Do you salt your food?	No	61.0	60.2	62.4	58.7	0.630
	Yes	39.0	39.8	37.6	41.3	
Do you eat sweets?	No	19.7	20.5	20.1	16.8	0.553
	Yes	80.3	79.5	79.9	83.2	
Do you eat fast food?	No	30.7	32.1	31.0	27.0	0.448
	Yes	69.3	67.9	69.0	73.0	
How much water do you consume daily?	Less than 1 l/day	17.6	21.1	14.9	18.7	0.171
	Approximately 1.5 l/day	52.8	52.1	54.1	50.8	
	More than 2 l/day	29.5	26.9	31.0	30.5	
Do you pay attention to the calorie content of your meals?	Always	18.7	22.2	18.2	13.3	0.106
	Never	31.0	31.0	31.3	30.3	
	Sometimes	50.3	46.8	50.6	56.4	
Do you read food labels when you shop?	Always	26.9	31.1	26.2	20.7	0.109
	Never	13.3	11.9	13.3	16.0	
	Sometimes	59.9	56.9	60.6	63.3	
Do you eat while watching TV, working on the computer?	Always	8.1	6.6	7.4	13.2	**0.022**
	Never	17.4	15.5	19.7	14.8	
	Sometimes	74.4	77.9	73.0	72.0	

*p*—*p* value for chi-square test result. Bold values denote statistical significance at the *p* < 0.05 level.

**Table 3 ijerph-19-03069-t003:** Characteristics of soldiers’ physical activity as a function of their BMI category (*n* = 1096).

Question	All		BMI 18.5–24.9		BMI 25–29.9		BMI ≥ 30		*p*
	X (SD)	Me	X (SD)	Me	X (SD)	Me	X (SD)	Me	
Number of days in the last week (7 days) on which soldiers performed moderate physical activity	2.81 (1.98)	3.00	3.01 (1.98)	3.00	2.90 (1.98)	3.00	2.14 (1.81)	2.00	**<0.001**
Time spent in front of a computer/tablet per day (h)	3.71 (3.05)	3.00	3.62 (2.76)	3.00	3.57 (3.09)	2.50	4.31 (3.41)	3.00	0.045
Time spent in front of TV per day (h)	1.55 (1.28)	1.00	1.40 (1.22)	1.00	1.59 (1.30)	1.50	1.76 (1.31)	2.00	**0.003**

X—mean value, SD—standard deviation, Me—median, *p*—*p* value for Kruskal–Wallis test result. Bold values denote statistical significance at the *p* < 0.05 level.

**Table 4 ijerph-19-03069-t004:** The prevalence of obesity in the childhood of the surveyed soldiers and their parents depending on the current BMI category of the soldiers.

Question	Response	All (%)	BMI 18.5–24.9 (%)	BMI 25–29.9 (%)	BMI ≥ 30 (%)	*p*
Were you of normal weight as a child?	Yes	76.3	76.7	77.3	72.3	**<0.001**
	No, I was overweight	11.4	5.6	12.0	20.7	
	No, I was obese	2.3	1.1	2.2	4.8	
	No, I was underweight	10.1	16.7	8.5	2.1	
Were you of normal weight at the age of 18?	Yes	82.2	82.8	82.9	79.3	**<0.001**
	No, I was overweight	8.7	4.7	8.8	16.0	
	No, I was obese	1.3	0.0	1.8	2.1	
	No, I was underweight	7.8	12.5	6.4	2.7	
Was your mother of normal weight	Yes	62.7	72.0	61.8	47.3	**<0.001**
	No, she was overweight	30.7	24.1	31.8	40.3	
	No, she was obese	5.3	2.8	5.1	10.8	
	No, she was underweight	1.3	1.1	1.3	1.6	
Was your father of normal weight	Yes	66.7	69.6	68.5	56.2	**0.010**
	No, he was overweight	26.3	25.1	23.9	35.7	
	No, he was obese	4.6	2.5	5.2	7.0	
	No, he was underweight	2.3	2.8	2.4	1.1	

*p*—*p* value for chi-square test result. Bold values denote statistical significance at the *p* < 0.05 level.

**Table 5 ijerph-19-03069-t005:** Risk factors for obesity in professional soldiers of the Polish Armed Forces (OR). Please note that the overweight group is not included in these analyses (further explanation in the text).

Risk Factor	Category	% Soldiers with BMI 18.5–24.9	% Soldiers with BMI ≥ 30	OR	95% CI
Sleep duration up to 6 h	No (7 h or more)	73.3	26.7	1	1.19–2.54
	Yes (up to 6 h)	61.1	38.9	1.74	
Eating when experiencing negative emotions, bad mood and well-being	No (I forget to eat or it does not affect my eating behavior)	70.5	29.5	1	2.12–6.22
	Yes (I reach for food more often)	41.5	58.5	3.64	
Excessive maternal weight	No	74.5	25.5	1	1.98–4.14
	Yes	50.5	49.5	2.86	
Excessive father’s weight	No	70.4	29.6	1	1.23–2.57
	Yes	57.1	42.9	1.78	
Exercise less than 3 days per week	No (3 or more)	74.6	25.4	1	1.59–3.27
	Yes (up to 2 days)	56.3	43.7	2.28	
Time spent in front of the TV more than 2 h a day	No (up to 2 h)	70.9	29.1	1	1.07–2.23
	Yes (2 h or more)	61.3	38.7	1.54	
Age over 40 years	No (up to 39 years)	76.3	23.7	1	3.04–6.55
	Yes (40 and over)	41.9	58.1	4.46	

**Table 6 ijerph-19-03069-t006:** Logistic regression results for statistically significant obesity risk factors among surveyed soldiers.

Variable	B	Standard Error	*p*	OR	95% CI
Eating when experiencing negative emotions, bad mood and well-being	0.994	0.261	**<0.001**	2.70	1.62–4.51
Excessive maternal weight	0.968	0.205	**<0.001**	2.63	1.76–3.94
Moderate exercise less than 3 days per week	0.599	0.204	**0.003**	1.82	1.22–2.72
Age over 40 years	1.315	0.212	**<0.001**	3.72	2.46–5.65
Constant	−1.977	0.193	**<0.001**		

B—regression coefficient, *p*—*p* value. Bold values denote statistical significance at the *p* < 0.05 level.

## Data Availability

Data sharing is not applicable to this article.

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
