# Peer review of "Assessment of Risk Factors for Development of Overweight and Obesity among Soldiers of Polish Armed Forces Participating in the National Health Programme 2016–2020"

_ijerph, 2022, doi:10.3390/ijerph19053069_

Round 1
Reviewer 1 Report
The manuscript submitted for publication to IJERPH by Gaździńska et al., titled: "Assessment of risk factors for development of overweight and obesity among soldiers of Polish Armed Forces participating in the National Health Programme 2016-2020" evaluated the risk factors leading to obesity/overweight of military personnel of Polish Armed Forces.
The manuscript is well written and is nicely organized. It flows and reads well and addresses an interesting topic that is typically understudied.
The reviewer would like to offer the following points for consideration from the authors:
- Consider providing any background information available for the state of health in the Polish Armed Forces.
- Is there any background information of health challenges in the military personnel field in general?
- Did the authors consider level and type of physical activity, in addition to frequency?
- Did the authors consider diet (type and quality)?
- BMI does not have units (despite WHO designation) kg/m2 denotes weight over surface area. Being an index BMI is unit-less.
- Consider using the term participants instead of subjects.
- How was the questionnaire developed? Was it validated?
Author Response
Response to Reviewer 1 Comments
Point 1: Consider providing any background information available for the state of health in the Polish Armed Forces.
Response 1: The required information was added on page 2.
Point 2: Is there any background information of health challenges in the military personnel field in general?
Response 2: The information was supplemented in the introduction on page 2.
Point 3: Did the authors consider level and type of physical activity, in addition to frequency?
Response 3: Yes, but it was very short. Only one question in the survey asked for „number of days during the last week (7 days) when the respondent did moderate physical exercise”. The response to this question turned out to be a significant factor predicting BMI≥ 30 (page 4).
Point 4: Did the authors consider diet (type and quality)?
Response 4: Only a small part of the participants of this study filled Food Frequency Questionnaires. Therefore, the findings will be presented in a separate manuscript.
Point 5: BMI does not have units (despite WHO designation) kg/m2 denotes weight over surface area. Being an index BMI is unit-less.
Response 5: The kg/m2 units have been removed from the manuscript.
Point 6: Consider using the term participants instead of subjects.
Response 6: The term “subjects” was changed to „participants” throughout the manuscript.
Point 7: How was the questionnaire developed? Was it validated?
Response 7: The questionnaire was developed in 2017 at the Laboratory of Dietetics and Obesity Treatment at the Military Institute of Aviation Medicine in Warsaw, based on the current literature review on obesity risk factors by an experienced research team. The questionnaire has been used in research since 2017. To date, it has been completed by more than 2000 subjects. The questions in the survey were approved by experts from the Polish Department of Military Health Services, who oversaw the implementation of the National Health Programme, before the start of the study.
Reviewer 2 Report
The manuscript by Grazdzinska et al. examines the prevalence of overweight and obesity in the Polish armed forces and focuses on evaluating different lifestyle/behavioral risk factors for developing these conditions in this specific population. The article is generally well written and easily understandable.
Some comments:
- Did the authors perform any nutritional assessments (24-h recalls or FFQ)? It would be interesting to see the estimated total energy and main nutrient intakes in the subgroups of the studied population.
- The authors should consider moving the part of the text describing the regression model (Lines 206-216) to the Material and Methods section.
-The authors could have used the body composition analyzer to get the information on the muscle and fat mass to overcome the BMI-related obesity overdiagnosis in this specific group. Perhaps, the authors should consider mentioning this in the limitation section.
- In the conclusion section, the authors should consider underlying the novelty of the obtained results and their significance.
- The authors should perhaps consider providing a figure summarizing the obtained results. There are a lot of tables, and a summary figure could more easily focus the reader’s attention on the main results. Alternatively, some of the data could be presented graphically.
Author Response
Response to Reviewer 2 Comments
Point 1: Did the authors perform any nutritional assessments (24-h recalls or FFQ)? It would be interesting to see the estimated total energy and main nutrient intakes in the subgroups of the studied population.
Response 1: Only a small part of the participants of this study completed Food Frequency Questionnaires. Therefore, the findings will be presented in a separate manuscript.
Point 2: The authors should consider moving the part of the text describing the regression model (Lines 206-216) to the Material and Methods section.
Response 2: It was moved to the Material and Methods section on page 3.
Point 3: The authors could have used the body composition analyzer to get the information on the muscle and fat mass to overcome the BMI-related obesity overdiagnosis in this specific group. Perhaps, the authors should consider mentioning this in the limitation section.
Response 3: We agree with the reviewer. However, to be consistent with current literature and to allow comparisons, we keep using the BMI to “diagnose” obesity. Anyway, it was demonstrated in other studies that soldiers with BMI in the obese range also have increased amount of body fat. This limitation is mentioned now in the limitations section on page 10.
Point 4: In the conclusion section, the authors should consider underlying the novelty of the obtained results and their significance.
Response 4: We have highlighted the conclusion that current efforts to prevent overweight and obesity among soldiers are insufficient. Furthermore we provided several recommendation based on our findings. They are now provided in the Conclusion section on page 10.
Point 5: The authors should perhaps consider providing a figure summarizing the obtained results. There are a lot of tables, and a summary figure could more easily focus the reader’s attention on the main results. Alternatively, some of the data could be presented graphically.
Response 5:
It is standard practice to present the results of logistic regression in the form of a table and model, which has been done in this paper. The table summarising the results obtained is Table 6.
Reviewer 3 Report
This is an interesting study performed among a big sample of Polish Army soldiers which is using a sound methodology to identify factors associated with obesity among them.
There are some issues which require further clarifications
- In the abstract and methodology authors mention now that they are looking at environmental factors, which in fact they are also looking into several other factors (biological, behavioural)
- The information about data collection and analyses ( inclusive formula and approaches used for regression analyses) should be all included in the Methodology, not in the Results
- The methodology should include a subsection like Instruments for data collection where the instruments should be described
- Tables depicting information about the study sample and BMI should be included in the Results section, not Methodology
- The statistical analyses should be better explained ( for instance the type of regression analyses, dependent and independent variables should be better described)
- The tables depicting the results of regression analyses should be reorganized to show these results, not prevalences again (this information is depicted in previous tables)
- It would be good to have in the Conclusions some indications/recommandations for future health programmes/initiatives and/or research
Author Response
Response to Reviewer 3 Comments
Point 1: In the abstract and methodology authors mention now that they are looking at environmental factors, which in fact they are also looking into several other factors (biological, behavioural).
Response 1: Now we specify in the abstract that only selected factors are evaluated. The selection was based on previous literature and was also extended to include other factors. The selection is described in the Introduction.
Point 2: The information about data collection and analyses (inclusive formula and approaches used for regression analyses) should be all included in the Methodology, not in the Results.
Response 2: The relevant paragraph was moved to the Material and Methods section on page 3.
Point 3: The methodology should include a subsection like Instruments for data collection where the instruments should be described.
Response 3: A section 2.3 “Instruments for data collection” was created and it included the description the questionnaire used.
Point 4: Tables depicting information about the study sample and BMI should be included in the Results section, not Methodology.
Response 4: Table 1 has been moved to the Results section.
Point 5: The statistical analyses should be better explained (for instance the type of regression analyses, dependent and independent variables should be better described).
Response: It was corrected (page 3 and page 4).
Point 6: The tables depicting the results of regression analyses should be reorganized to show these results, not prevalences again (this information is depicted in previous tables).
Response 6: Table 5 presents odds ratios for selected risk factors for obesity among soldiers, while Table 6 presents only statistically significant risk factors after introducing multivariate regression into the model.
Point 7: It would be good to have in the Conclusions some indications/recommendations for future health programmes/initiatives and/or research.
Response 7: We have added the recommendations to the Conclusions section.
Round 2
Reviewer 1 Report
The authors have made a reasonable effort to address reviewer's comments. Proofreading is recommended.
Reviewer 3 Report
The authors answered to all comments and improved the new version of the article.